# Changes in *Aquaporin 1*, *5* and *9* Gene Expression in the Porcine Oviduct According to Estrous Cycle and Early Pregnancy

**DOI:** 10.3390/ijms21082777

**Published:** 2020-04-16

**Authors:** Damian Tanski, Agnieszka Skowronska, Maciej Eliszewski, Leszek Gromadzinski, Bartosz Kempisty, Mariusz T. Skowronski

**Affiliations:** 1Department of Animal Anatomy and Physiology, University of Warmia and Mazury in Olsztyn, 10-719 Olsztyn, Poland; 2Department of Human Histology and Embryology, School of Medicine, University of Warmia and Mazury in Olsztyn, 10-752 Olsztyn, Poland; 3Department of Human Physiology and Pathophysiology, School of Medicine, University of Warmia and Mazury in Olsztyn, 10-752 Olsztyn, Poland; agnieszka.skowronska@uwm.edu.pl; 4Department of Gynecology and Obstetrics, School of Medicine, University of Warmia and Mazury in Olsztyn, 10-752 Olsztyn, Poland; elisz@wp.pl; 5II Department of Cardiology and Internal Medicine, Collegium Medicum, School of Medicine, University of Warmia and Mazury, 10-719 Olsztyn, Poland; leszek.gromadzinski@uwm.edu.pl; 6II Department of Cardiology and Internal Medicine, University Clinical Hospital in Olsztyn, Warszawska 30, 10-082 Olsztyn, Poland; 7Department of Histology and Embryology; Poznan University of Medical Sciences, 61-701 Poznan, Poland; bkempisty@ump.edu.pl; 8Department of Anatomy, Poznan University of Medical Sciences, 61-701 Poznan, Poland; 9Department of Obstetrics and Gynecology, University Hospital and Masaryk University, 602 00 Brno, Czech Republic; 10Department of Veterinary Surgery, Institute of Veterinary Medicine, Nicolaus Copernicus University in Torun, 87-100 Torun, Poland; 11Department of Basic and Preclinical Sciences, Institute of Veterinary Medicine, Nicolaus Copernicus University in Torun, 87-100 Torun, Poland

**Keywords:** aquaporins, oviduct, estrous cycle, pregnancy, pig

## Abstract

Aquaporins (AQPs) are a group of small, integral membrane proteins which play an important role in fluid homeostasis in the reproductive system. In our previous study, we demonstrated AQP1, 5 and 9 protein expression and localization in the porcine oviduct. The presence of these isoforms could suggest their role in the transport of the ovum to the uterus by influencing the epithelial cells’ production of oviductal fluid. The aim of this study was to evaluate the expression of *AQP1*, *AQP5* and *AQP9* in the infundibulum, ampulla and isthmus in the porcine oviduct during the estrous cycle (early luteal phase, days 2–4, medium luteal phase, days 10–12, late luteal phase days 14–16, follicular phase days 18–20) and pregnancy (period before implantation, days 14–16 and after the implantation, days 30–32) using the Real-Time PCR technique. As clearly demonstrated for the first time, *AQP1*, *5*, and *9* gene expression is influenced by the estrus cycle and pregnancy. Furthermore, expression of AQPs in the porcine oviduct may provide the physiological medium that sustains and enhances fertilization and early cleavage-stage embryonic development. Overall, our study provides a characterization of oviduct AQPs, increasing our understanding of fluid homeostasis in the porcine oviduct to successfully establish and maintain pregnancy.

## 1. Introduction

Aquaporins (AQPs) belong to a family of intrinsic membrane proteins that act as selective channels for water and for solutes such as glycerol and urea. They play an important role in fluid homeostasis and the control of epithelial cell volume; furthermore, expressions of AQPs have been proven in the reproductive system. The oviduct is the place where fertilization and early embryonic development happens. The oviduct transports the ovum towards the uterus, but the mechanism responsible for this is still unknown. Pauerstein and Eddy [1] theory, called ‘tube locking’ says that the ovum may be delayed in the oviduct by muscle contraction, isthmic oedema [2,3] or vascular distension [4]. The porcine oviduct and its secretions through the late follicular stage and the first four days of the estrous cycle provide a proper environment that facilitates the establishment of pregnancy. Under the influence of ovarian steroid hormones, the oviduct microenvironment undergoes significant morphological, physiological and biochemical changes [5,6]. This microenvironment is settled by oviductal fluid changes in quality and quantity in response to fluctuations of the serum levels of the hormones during the estrous cycle [7]. Previously, we reported that aquaporins 1, 5 and 9 exist in the porcine oviduct in different stages of the cycle and early pregnancy under physiological conditions [8,9]. They seem to be very important for the normal oviductal functions to maintain fluid homeostasis by influencing the epithelial cells’ production of oviductal fluid, which provides the physiological medium for fertilization and early embryonic development. A few studies have also evaluated the expression of AQPs in the oviduct, with special attention to AQP1, AQP9 and AQP5. These findings indicate that AQPs could, therefore, play an important role in regulating ovum transport in the fallopian tube by altering the luminal diameter and rapidly increasing the movement of water through AQP1 into the smooth muscle cells, leading to muscle swelling and causing the shutdown of the lumen, thus ‘locking’ the tube [10]; and by hormonal regulation of the type and quantity of water channels in the epithelium, which might control water transport in the oviductal lumen [11]. Furthermore, recent studies indicate that AQP5 specific to non-ciliated cells is activated during sexual maturation, supporting fluid homeostasis in the mouse oviduct [12]. 

Nevertheless, the physiological significance of aquaporin-mediated water transport in the porcine oviduct is still very limited. To understand oviductal fluid homeostasis and to extend our knowledge, the aim of the present study was to investigate the expression of *AQP1*, *AQP5* and *AQP9* by Real-Time RT-PCR (Table 1).

## 2. Results

Gene expression of *AQP1*, *5* and *9* in parts of the porcine oviduct (infundibulum, ampulla and isthmus) during the estrous cycle and early pregnancy was found.

The expression of *AQP1*, *AQP5* and *AQP9* mRNA in the parts of the porcine oviduct at distinct stages of the estrous cycle is shown and summarized in Figure 1A–I. The expression of *AQP1* mRNA did not change significantly in the infundibulum but increased (*p* < 0.05) in the ampulla on days 10–12 and 18–20 (Figure 1B) and in the isthmus on days 14–16 and 18–20 (Figure 1C). The expression of *AQP5* mRNA increased only in the isthmus on days 14–16 and 18–20 (Figure 1F). The expression of *AQP9* mRNA did not change in the parts of the oviduct. 

The expression of the studied aquaporins’ mRNA in the parts of the porcine oviduct at different stages of early pregnancy (days 14–16 and 30–32) is shown and summarized in Figure 2A–I. The expression of *AQP1* mRNA significantly changed only (*p* < 0.05) in the isthmus on days 30–32 (Figure 2C), but also observed a trend in the infundibulum (*p* < 0.1) (Figure 2A). The expression of *AQP5* mRNA did not change significantly but observed a trend (*p* < 0.1) in the infundibulum and the ampulla on days 30–32 (Figure 2D,E). 

Figure 3 compares the expression of AQP1, 5 and 9 between the mid-luteal and late-luteal phase of the estrous cycle (days 10–12 and 14–16), in which corpus luteum activity is high and similar to that noted before implantation and at the end of implantation during pregnancy (days 14–16 and days 30–32). Hence, these phases of the cycle were control groups that represented the tissues of non pregnant animals. Similar analysis was previously performed by other authors [9,17]. The expression of *AQP1*, *AQP5* and *AQP9* mRNA in the parts of the porcine oviduct (infundibulum, ampulla, isthmus) at different stages of the estrous cycle (days 10–12 and 14–16) compared to early pregnancy (days 14–16 and 30–32) did not change significantly. The expression of *AQP5* mRNA increased significantly only in the isthmus on days 14–16 of the estrous cycle relative to early pregnancy (days 30–32) (Figure 3D), and a trend was observed (*p* < 0.1) in the isthmus during the estrous cycle on days 10–12 (Figure 3C).

## 3. Discussion

In this study, gene expression of *AQP1*, *5* and *9* in parts of the porcine oviduct (infundibulum, ampulla and isthmus) during the estrous cycle and early pregnancy was examined. The gene expression of *AQP1* significantly changed in the ampulla on days 10–12 and 18–20 and in the isthmus on days 14–16 and 18–20 of the estrous cycle (Figure 1B,C). The expression of AQP5 mRNA increased only in the isthmus on days 14–16 and 18–20 of the estrous cycle (Figure 1F). During early pregnancy, the expression of AQP1 mRNA significantly changed in the isthmus on days 30–32 (Figure 2C). Next, we did a comparison of AQP1, 5 and 9 mRNA expression in the parts of the oviduct during the estrous cycle and pregnancy. The AQP5 mRNA expression was only significantly higher on days 14–16 of the estrous cycle when compared to pregnancy (days 30–32), Figure 3D. This result ties well with previous studies, wherein we found that in the oviduct of cyclic and pregnant gilts, the expression of AQP1, AQP5 and AQP9 proteins significantly increased on days 2–4 and 18–20 of the estrous cycle [9].

The first information about the presence of AQPs in the reproductive system comes from Li and coworkers [18]. In the following years, the studies revealed the presence of AQPs in different structures of the reproductive system of the human, rat and mouse [19]. Moreover, we showed the presence of AQP1, AQP5 and AQP9 proteins in the ovary, oviduct and uterus of pigs [8]. Our previous studies clearly indicated that the expression of AQP1, AQP5 and AQP9 proteins in the pig ovarian follicles and oviduct, as well as in the endometrium and myometrium, is affected by the estrous cycle and early pregnancy [9,20,21]. Taken together, these results suggest that estrogens and progesterone (P4) are involved in the expression of AQPs in the reproductive system of the pig. 

In mammals, embryo transport is an interactive process between the embryos and the oviduct. The three major elements regulating this transport are as follows: the beating of the ciliated epithelia, oviductal fluid flow and oviductal muscle contraction. Estrogen (E2) increases embryo transport by the acceleration of muscle contraction and fluid secretion as well as the ciliary beat frequency (CBF) of the oviduct while P4 acts opposite to this transport [22]. Beside E2 and P4, prolactin (PRL) plays a major role in fluid production in the oviduct [22]. We recently found that PRL increased AQP5 mRNA and protein expression in the granulosa (GC) and theca cells (TC) obtained from medium (MF) and large follicles (LF) and in cocultures of these cells [23]. Prolactin had a stimulatory effect on AQP1 protein expression in the coculture of the abovementioned cells [23]. Moreover, we indicated the functional AQPs expression in these cells using swelling assay with an AQPs blocker, because a significant increase in the cell volume in hypotonic conditions treated with PRL was demonstrated when compared to the control group. We concluded that AQP1 may be implicated in follicle development and in cell proliferation and migration [24]. Aquaglyceroporin-3 simplified water and glycerol transport in epidermal cell migration and proliferation [25]. Moreover, E2 caused the proliferation of secretory cells and differentiation into ciliated cells in the porcine oviduct [26]. Estradiol by activation of the epidermal growth factor (EGF) system, vascular endothelial growth factor (VEGF) and the fibroblast growth factor (FGF) system may influence the proliferation and differentiation of the porcine oviductal and endometrial cells during the estrous cycle and pregnancy [27,28].

Ovarian hormones have a stimulatory effect on the oviductal muscle contraction by both direct effects and indirect effects via the induction of prostaglandins (PGs) [21]. The results performed by Skowronska et al. [19] indicated specific patterns of the *AQP1* and *AQP5* gene and protein expression in response to E2, P4 and arachidonic acid (AA; substrate for prostaglandins synthesis) in the cells of the porcine endometrium and the myometrium during the estrus cycle and pregnancy. Furthermore, this research demonstrated that ovarian hormones and AA cause translocation of AQP5 from apical to the basolateral plasma membrane of the epithelial cells, which might affect the transcellular water movement (through the epithelial cells) between the uterine lumen and the blood vessels via AQP1. Branes et al. [11] and Gannon et al. [10] found AQP5, 8 and 9 expression in the rat epithelial cells and AQP1 in the membranes of smooth muscle cells of the oviduct, respectively. These authors suggested that by increasing water transport into the smooth muscle cells of the oviduct, AQPs caused muscle swelling and the shutdown of the lumen that facilitated ovum movement towards the uterus: the so called “tube locking” hypothesis [10]. Other authors indicated the expression of AQPs in the muscle cells of the female reproductive system. For example, Helguera et al. [29] and Girotti and Zingg [30] reported varied AQP5 and AQP8 expression in rat myometrium by microarray and qRT-PCR during pregnancy. Wanggren et al. [31] showed that oviductal muscular contractions in the human increased after treatment with prostaglandin F2α and prostaglandin E2. The contractions decreased after prostaglandin E1, P4, levonorgestrel, mifepristone, oxytocin and hCG. The study by Skowronski [20] confirms AQP5 expression in the smooth muscle cells of the pig uterus. Moreover, we found that oviductal muscles exhibited AQP5 expression and that AQP1 was present in the vessels [9].

Issues associated with the acquisition of developmental capacity by cumulus oocyte complexes (COCs) in females still pose a challenge for modern researchers. In most species of mammals, the oocyte matures within the ovarian follicle and during the ovulation period it is mainly in the stage of metaphase II (MII) meiosis. In light of studies on animals, it is assumed that an important role in these mechanisms can be played by connexins (Cxs) [32,33,34] and aquaporins (AQPs) [8,19,23,35]. Gap junctions appearing in oocyte-cumulus oophorus cell complexes are composed of protein Cxs [36]. Investigations carried out over the last twenty years show that these proteins’ deficiency may result in developmental disturbances of oocytes and their infertility [37,38,39]. 

McConnell et al. [40] showed that the rate of water movement into the antral cavity of rat follicles is primarily transcellular, which may be mediated by AQP7, AQP8 and AQP9 in granulosa cells. In particular, the expression of AQP7 and AQP9 (aquaglyceroporins), which in addition to water can transport glycerol, urea and other small particles suggesting their involvement in the rapid transport of small neutral molecules, may also be important in the development of ovarian follicles [40]. For example, AQP7 and AQP9 can provide a path for androgen substrates for the production of estrogen in granulosa cells. A similar meaning can be attributed to *AQP3* and *AQP7* in mice [41,42]. Understanding the expression and function of AQPs in the cells of ovarian follicles can lead to a better understanding of mechanisms regulating the oocyte maturation processes and build up their capacity for fertilization and normal development of the embryo. A measurable effect of knowledge on these mechanisms may be the effectiveness of the pig oocyte cryopreservation. It has been found that the increased expression of *AQP3* in mouse oocytes improves water and glycerol permeability and the survival of frozen ova [41]. The authors suggest that changes in the expression of AQPs in oocytes may affect their survival after freezing. In contrast, research performed by Meng et al. [42] demonstrated that controlled ovarian stimulation reduces AQP3 mRNA expression in mouse oocytes, which may be one of the reasons for poor survival of frozen oocytes. To date, methods successful in protecting sperm and embryos have proven to be unreliable for oocytes. In addition, difficulties in the cryopreservation of oocytes are a major factor limiting the free movement of genetic material and reproduction of animals [41]. Understanding the expression and function of AQPs in the cells of ovarian follicles can lead to a better understanding of mechanisms regulating the oocyte maturation processes and build up their capacity for fertilization and normal development of the embryo. 

It has been suggested that oocyte surrounding somatic cells, such as cumulus oophorus cells (COCs), granulosa cells and oviductal epithelial cells (OECs), may play a significant role in regulating the maturation stages essential for the oocyte. Recent research shows that adhesion has a great impact on the post-ovulatory journey of the oocyte through the oviduct. It was found that in cows the interaction between COCs and OECs starts immediately after the COCs’ entry to the oviductal ampulla [43], promoting hyperactivation of sperm bound in the isthmus and its counter-current movement towards the ampulla [44]. The study, performed by Budna-Tukan et al. [45], found and focused on genes belonging to the “biological adhesion” gene ontology biological processes. The authors suggested that increased expression of these genes may be important for the secretion of substances that influence the oviductal environment, gametes’ adhesion, fertilization and embryo transport [45].

## 4. Materials and Methods 

### 4.1. Experimental Animals and Collection of Oviduct Tissue

All experiments were performed in accordance with the principles and procedures of the Animal Ethics Committee (number 32/2012), University of Warmia and Mazury in Olsztyn, Poland. Tissue samples were taken from mature cross-bred gilts (Large White × Polish Landrace) descended from the private breeding farm, at the early (days 2–4; *n* = 5), middle (days 10–12; *n* = 5) and late (days 14–16; *n* = 5) stage of the luteal phase and the stage of the follicular phase (days 18–20; *n* = 5) of the estrous cycle as well as on days 14 to 16 (*n* = 5) and 30 to 32 (*n* = 5) of gestation (the onset and the end of implantation process, respectively). The gilts were observed daily for estrous behavior in the presence of a boar. The day of onset of the second estrous was marked as day 0 of the estrous cycle. The phase of the estrous cycle was also confirmed based on the characteristic morphology of the ovaries [46]. Regarding pregnant pigs, natural insemination was performed on days 1 to 2 of the estrous cycle. After slaughter, the pig reproductive tracts were transported to the laboratory on ice. Pregnancy was confirmed by the presence and morphology of conceptuses [47].

### 4.2. Preparation of Oviduct Slices

Sections of the parts of the oviducts collected from the pigs were sliced to three sections: infundibulum, ampulla and isthmus. Furthermore, oviduct tissue explants were snap-frozen in liquid nitrogen (for RNA extraction) and stored at −80 °C until further use.

### 4.3. RNA Isolation

Total RNA was extracted, using the total RNA Mini kit (A&A Biotechnology, Gdansk, Poland) according to the manufacturer’s protocol, from oviduct explants collected after being snap-frozen. Total RNA quality and quantity were determined with spectrophotometry. (Infinite^®^ 200 PRO NanoQuant, Tecan, Switzerland).

### 4.4. cDNA Synthesis and Quantitative Real-Time Polymerase Chain Reaction Analysis

Total RNA samples were transcribed to cDNA using a TransScriba Kit (A&A Biotechnology, Gdansk, Poland). Real-Time PCR was performed in triplicate for each sample using a AriaMx Real-Time PCR System (Agilent Technologies, Santa Clara, CA, USA) and a SYBR^®^Green PCR Master Mix (Life Technologies, Grand Island, NY, USA). Real-Time PCR reaction included 12.5 μL SYBR Green PCR master mix, 1 μM forward and reverse primers each and reverse transcribed cDNA (2 μL of diluted RT product) supplemented with water to a volume of 25 μL. The conditions of the thermal cycling for each gene were: initial denaturation for 10 min at 95 °C, denaturation for 15 sec at 95 °C, primer annealing for 1 min at 60 °C. Specific primers for *AQP1*, *AQP5* and *AQP9* (Table 2.) were designed with the PrimerQuest Tool (Integrated DNA Technologies, Inc., Coralville, USA), and their specificities were confirmed by the comparison of their sequences with the sequence of *AQP1*, *AQP5* and *AQP9* deposited in a database, and the calculation of the statistical significance of the match was performed using the Basic Local Alignment Search Tool (BLAST). For the specificity control, non-template controls and dissociation curve analysis of the amplified products were used for each amplification. The specificity of amplifications was further validated with electrophoresis of the putative amplicons in a 2% agarose gel. Levels of gene expression were calculated using the ΔΔ C*t* method and normalized using the geometrical means of reference genes expression levels, *Cyclophilin* (*PPIA*) described by Lord et al. [48] and *18S rRNA* (Table 2.).

### 4.5. Statistical Analysis 

The data are presented as means ± SEM from five different observations. Differences between groups within each phase of the estrous cycle and early pregnancy were analyzed separately by one-way ANOVA followed by a least-significant differences post hoc test. Statistical analyses were performed using Statistica Software (StatSoft Inc., Tulsa, USA). Values for *p* < 0.05 were considered statistically significant.

## 5. Conclusions

The main conclusion that can be drawn is that *AQP1*, *5* and *9* gene expression in the porcine oviduct is affected by the estrus cycle and pregnancy. These results as well as our previous findings [9] may indicate that AQP1, AQP5 and AQP9 expression at the mRNA and protein level is important for maintaining an appropriate fluid environment for fertilization and embryonic development in the pig oviduct. However, future studies could investigate the association between AQPs and other genes/proteins; for example, connexins or genes belonging to “biological adhesion” ontology biological processes that influence the porcine oviductal environment during the estrous cycle and early pregnancy.

## Figures and Tables

**Figure 1 ijms-21-02777-f001:**
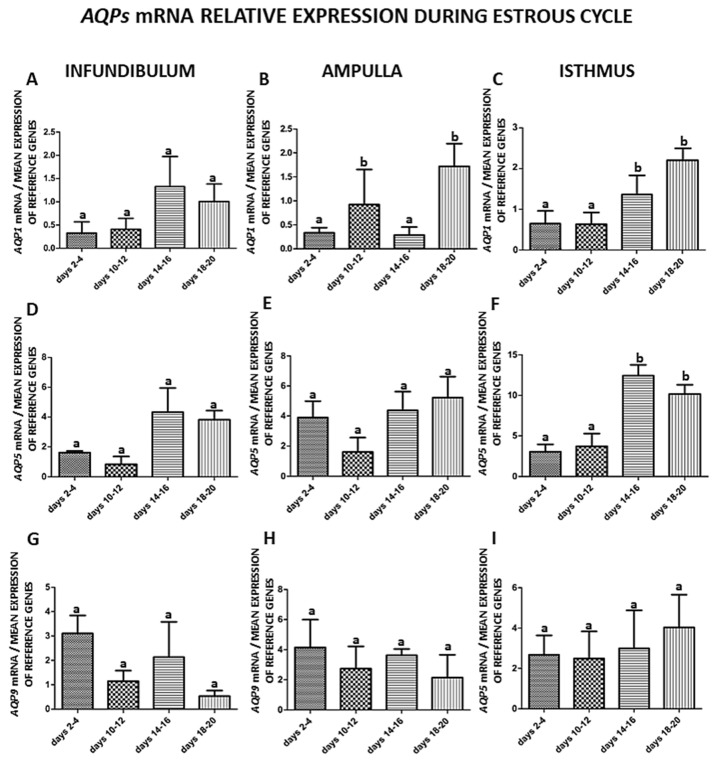
The expression patterns of *Aquaporin 1* (**A**–**C**)*, Aquaporin 5* (**D**–**F**) and *Aquaporin 9* (**G**–**I**) genes in porcine oviduct separated into infundibulum, ampulla and isthmus from the estrus cycle (days 2–4, 10–12, 14–16, 18–20). The samples were subjected to quantitative polymerase chain reaction (qPCR) analysis and normalized to the geometric mean of *18S RNA* and *Cyclophilin A* (*PPIA*) expression. Each of the graphs marked with a capital letter indicates a specific part of the oviduct. Values are expressed as means ± S.E.M from five individuals, each performed in triplicate (*p* < 0.05) (*n* = 5). Statistically significant differences between the part of the oviduct to day of the estrous cycle are indicated by different letters (a, b).

**Figure 2 ijms-21-02777-f002:**
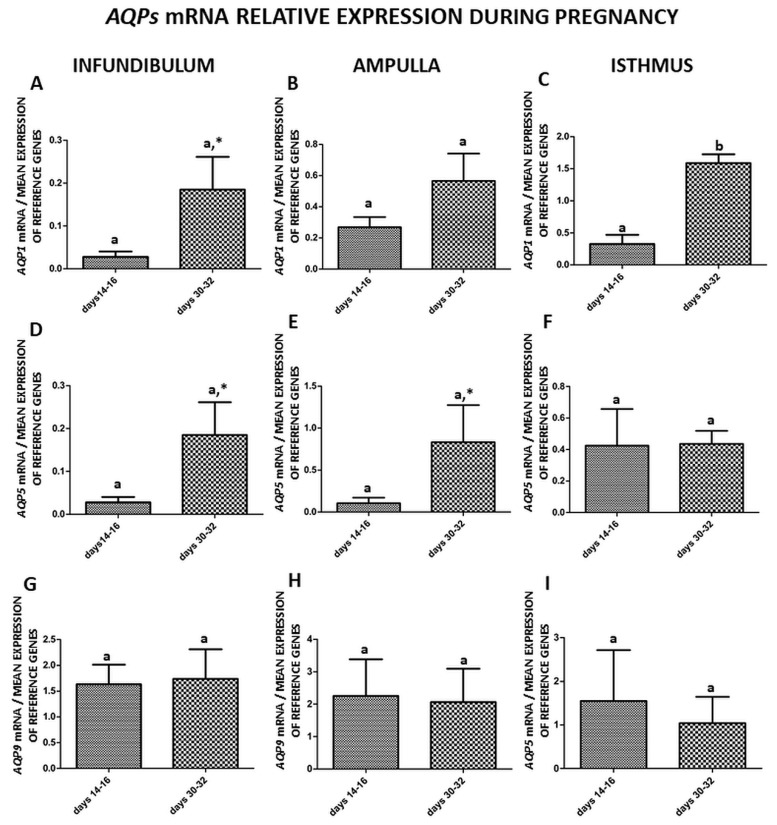
The expression patterns of *Aquaporin 1* (**A**–**C**), *Aquaporin 5* (**D**–**F**), and *Aquaporin 9* (**G**–**I**), genes in porcine oviduct separated into infundibulum, ampulla and isthmus from early pregnancy (days 14–16 and 30–32). The samples were subjected to qPCR analysis and normalized to the geometric mean of *18S RNA* and *PPIA* expression. Each of the graphs marked with a capital letter indicates a specific part of the oviduct. Values are expressed as means ± S.E.M from five individuals, each performed in triplicate (*p* < 0.05) (*n* = 5). Statistically significant differences between the part of the oviduct to day of the estrous cycle are indicated by different letters (a, b) and trends (*p* < 0.1) by *.

**Figure 3 ijms-21-02777-f003:**
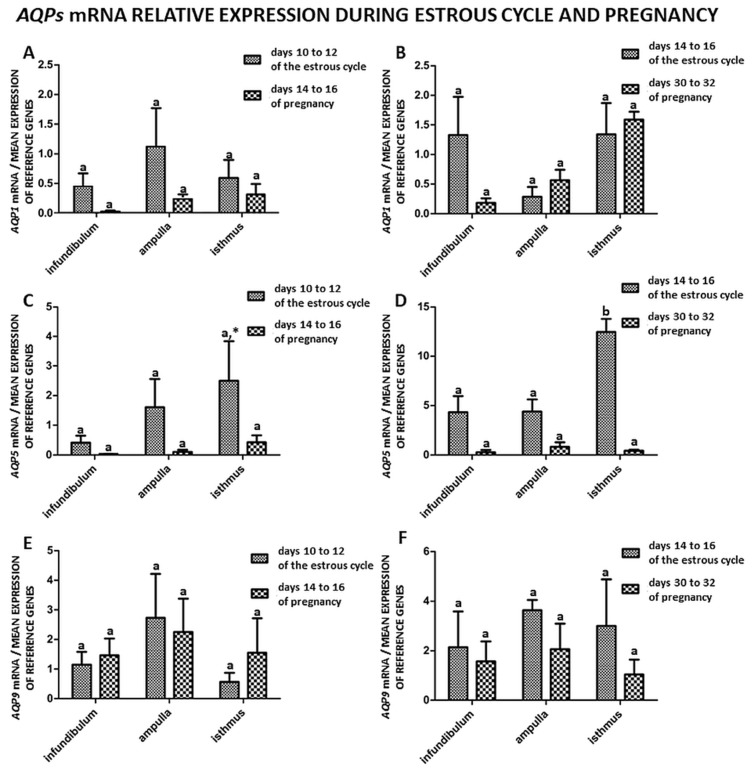
Comparison of the expression patterns of *AQP1*, *5*, *9* genes in porcine oviduct separated into infundibulum, ampulla and isthmus from the estrus cycle (days 10–12 and 14–16) and early pregnancy (days 14–16 and 30–32). (**A**,**C**,**E**) summarized comparison between medium luteal phase (days 10–12) and pregnancy (days 14-16). (**B**,**D**,**F**) summarized comparison between late luteal phase (days 14–16) and pregnancy (days 30–32). The samples were subjected to qPCR analysis and normalized to the geometric mean of *18S RNA* and *PPIA* expression. Each of the graphs marked with a capital letter indicates a specific part of the oviduct. Values are expressed as means ± S.E.M from five individuals, each performed in triplicate (*p* < 0.05) (*n* = 5). Statistically significant differences between the part of the oviduct from the estrous cycle and early pregnancy are indicated by different letters (a, b) and trends (*p* < 0.1) by *.

**Table 1 ijms-21-02777-t001:** Expression of aquaporin isoforms in oviductal tissues.

Aquaporin Isoforms	Species	Vessels	Muscles	Luminal Epithelium	Articles
AQP1	Rat	+		+	Gannon BJ. et al. [10]
	Pig	+			Skowronski MT et al. [8]
	Goat	+	+	+	Arrighi S et al. [13]
AQP2	Turkey			+	Zaniboni L. and Bakst MR. [14]
AQP3	Turkey			+	Zaniboni L. and Bakst MR. [14]
AQP4	Goat			+	Arrighi S et al. [13]
	Chicken	+	+	+	Socha JK. et al. [15]
AQP5	Rat			+	Branes MC et al. [11]
	Pig		+	+	Skowronski MT et al. [8]
	Mouse		+	+	Nah WH. et al. [12]
	Goat			+	Arrighi S. et al. [13]
AQP6	-	-	-	-	-
AQP7	-	-	-	-	-
AQP8	Rat			+	Branes MC. et al. [11]
AQP9	Turkey			+	Zaniboni L. and Bakst M.R. [14]
	Rat			+	Branes MC. et al. [11]
	Pig			+	Skowronski MT. et al. [8]
	Human			+	Ji YF et al. [16]
AQP10	-	-	-	-	-
AQP11	-	-	-	-	-
AQP12	-	-	-	-	-

**Table 2 ijms-21-02777-t002:** Forward and reverse primers sequences, amplicons length and GeneBank accession numbers of genes used during Real-Time PCR analysis.

Name of the Gene	Primer Sequence Forward/Reverse	Amplicon Length, bp	Accession Number
*Aquaporin 1* (*Aqp1*)	5′-CAGCGAGTTCAAGAAGAAG-3′ 5′-GCGACACCTTCACGTTATC-3′	161	NM_214454.1
*Aquaporin 5* (*Aqp5*)	5′-CTATGAGTCCGAGGAGGATT-3′ 5′-GCTTCGCTGTCATCTGTT-3′	147	NM_001110424.1
*Aquaporin 9* (*Aqp9*)	5′-TCTGGTGGATTCCTGTAGTG-3′ 5′-GGTTTGTCCTCCGATTGTTC-3′	130	NM_001112684.1
*Cyclophilin* (*PPIA*)	5′-GCACTGGTGGCAAGTCCAT-3′ 5′-AGGACCCGTATGCTTCAGGA-3′	71	XM_021078519.1
*18S ribosomal RNA* (*18S rRNA*)	5′-GGCTACCACATCCAAGGAAG-3′ 5′-TCCAATGGATCCTCGCGGAA-3′	149	AK393333.1

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
