# Peer review of "Changes in *Aquaporin 1*, *5* and *9* Gene Expression in the Porcine Oviduct According to Estrous Cycle and Early Pregnancy"

_ijms, 2020, doi:10.3390/ijms21082777_

Round 1

Reviewer 1 Report

The paper entitled : “Changes in aquaporin 1, 5 and 9 gene expression in the porcine oviduct according to estrous cycle and early pregnancy” addresses changes in mRNA expression of AQP1, AQP5 and AQP9 in the porcine oviduct during the estrous cycle and early pregnancy stages using qPCR. Their findings support some changes in the mRNA expression levels of these AQPs at different stages and in different parts of the oviduct.

Although qPCR data are informative in terms of gene expression, one cannot infer physiological/functional implications from such data.  As it stands, the study does not further the understanding of a possible role of AQP1, 5 and 9 in oviductal fluid homeostasis. However, the authors have prior publications where AQPs’ protein levels/labeling patterns have been investigated. A better integration of these with the present findings would strengthen the discussion/conclusion.

  • There is no mention of how the animals were sacrificed.
  • Fig 1A (Infundibulum) and Fig 1C (Isthmus) are identical!
  • Figs 3 and 4 are essentially a merge between Figs 1 and 2 with the aim of comparing expression levels of AQPs between the estrus cycle and pregnancy in Fig 3 and between the different parts of the oviduct during the estrus cycle and during pregnancy. This is redundant and doesn’t bring anything more than what is already illustrated in Figures 1 and 2.
  • Why are the values in Fig 3A (around 0.7) and Fig 1B (around 0.9) for AQP1 in the ampulla at 10-12 different?
  • Why are the values in the isthmus at 10-12 different between Fig 1C (<0.5) and 3A (>0.5)?

Author Response

Response to Reviewer 1 Comments

We are sending the revised manuscript of our paper entitled “Changes in aquaporin 1, 5 and 9 gene expression in the pig oviduct according to estrous cycle and early pregnancy.” with hope it will fulfil scientific requirements to be published in your Journal

Point 1: -There is no mention of how the animals were sacrificed.

Response 1: We have changed and improved the Materials and Methods section, including physiological status of the animals.

Point 2: -Fig 1A (Infundibulum) and Fig 1C (Isthmus) are identical!

-Why are the values in Fig 3A (around 0.7) and Fig 1B (around 0.9) for AQP1 in the ampulla at 10-12 different?

-Why are the values in the isthmus at 10-12 different between Fig 1C (<0.5) and 3A (>0.5)?

Response 2: During the preparation of the manuscript an error occurred. We apologize for that. We changed Figure 1C and 3A. See attached corrected Figures.  

Point 3: Figs 3 and 4 are essentially a merge between Figs 1 and 2 with the aim of comparing expression levels of AQPs between the estrus cycle and pregnancy in Fig 3 and between the different parts of the oviduct during the estrus cycle and during pregnancy. This is redundant and doesn’t bring anything more than what is already illustrated in Figures 1 and 2.

Response 3:

Although every Reviewers opinion is appreciated we do not partly agree with this one.

We removed Fig 4 from the manuscript. However, data in Fig 3 is very important to show the comparison between  the luteal phase of the estrous cycle and early pregnancy, periods when corpus luteum is active. This model of comparison is well known in literature and often used by authors.

Below, there are several, our and others, peer reviewed publication that such comparison was made:

  1. Skowronski MT, Skowronska A, Nielsen S. Fluctuation of aquaporin 1, 5, and 9 expression in the pig oviduct during the estrous cycle and early pregnancy. J Histochem Cytochem. 2011;59(4):419–427. doi:10.1369/0022155411400874
  2. Skowronski MT. Distribution and quantitative changes in amounts of aquaporin 1, 5 and 9 in the pig uterus during the estrous cycle and early pregnancy. Reprod Biol Endocrinol. 2010;8:109. Published 2010 Sep 9. doi:10.1186/1477-7827-8-109
  3. Skowronska A, Mlotkowska P, Eliszewski M, Nielsen S, Skowronski MT. Expression of aquaporin 1, 5 and 9 in the ovarian follicles of cycling and early pregnant pigs. Physiol Res. 2015;64(2):237–245.
  4. Smolinska, N.; Kiezun, M.; Dobrzyn, K.; Rytelewska, E.; Kisielewska, K.; Gudelska, M.; Zaobidna, E.; Bogus-Nowakowska, K.; Wyrebek, J.; Bors, K.; Kopij, G.; Kaminska, B.; Kaminski, T. Expression of Chemerin and Its Receptors in the Porcine Hypothalamus and Plasma Chemerin Levels during the Oestrous Cycle and Early Pregnancy.  J. Mol. Sci.201920, 3887
  5. Kaczynski P, Bauersachs S, Baryla M, et al. Estradiol-17β-Induced Changes in the Porcine Endometrial Transcriptome In Vivo. Int J Mol Sci. 2020;21(3):E890. Published 2020 Jan 30. doi:10.3390/ijms21030890
  6. Smolinska N, Kaminski T, Siawrys G, Przala J. Expression of leptin and its receptor genes in the ovarian follicles of cycling and early pregnant pigs. Animal. 2013;7(1):109–117. doi:10.1017/S1751731112001103
  7. Wojciechowicz B, Kotwica G, Kolakowska J, Franczak A. The activity and localization of 3β-hydroxysteroid dehydrogenase/Δ(5)-Δ(4) isomerase and release of androstenedione and progesterone by uterine tissues during early pregnancy and the estrous cycle in pigs. J Reprod Dev. 2013;59(1):49–58. doi:10.1262/jrd.2012-099
  8. Franczak A, Wojciechowicz B, Kotwica G. Transcriptomic analysis of the porcine endometrium during early pregnancy and the estrous cycle. Reprod Biol. 2013;13(3):229–237. doi:10.1016/j.repbio.2013.07.001
  9. Klainbart S, Slon A, Kelmer E, et al. Global hemostasis in healthy bitches during pregnancy and at different estrous cycle stages: Evaluation of routine hemostatic tests and thromboelastometry. Theriogenology. 2017;97:57–66. doi:10.1016/j.theriogenology.2017.04.023
  10. Niswender GD, Juengel JL, McGuire WJ, Belfiore CJ, Wiltbank MC. Luteal function: the estrous cycle and early pregnancy. Biol Reprod. 1994;50(2):239–247. doi:10.1095/biolreprod50.2.239
  11. Kiewisz, Jolanta; Krawczyński, Kamil; Lisowski, Paweł; Blitek, Agnieszka; Zwierzchowski, Lech; Zięcik, Adam J.; Kaczmarek, Monika M. Global gene expression profiling of porcine endometria on days 12 and 16 of the estrous cycle and pregnancy. Theriogenology 2014, 82(6): 897-909 (doi: 10.1016/j.theriogenology.2014.07.009).
  12. Przygrodzka E, Witek KJ, Kaczmarek MM, Andronowska A, Ziecik AJ. Expression of factors associated with apoptosis in the porcine corpus luteum throughout the luteal phase of the estrous cycle and early pregnancy: their possible involvement in acquisition of luteolytic sensitivity. Theriogenology. 2015;83(4):535–545. doi:10.1016/j.theriogenology.2014.10.016
  13. Korzekwa AJ, Bah MM, Kurzynowski A, Lukasik K, Groblewska A, Skarzynski DJ. Leukotrienes modulate secretion of progesterone and prostaglandins during the estrous cycle and early pregnancy in cattle: an in vivo study. Reproduction. 2010;140(5):767–776. doi:10.1530/REP-10-0202
  14. Guzeloglu A, Atli MO, Kurar E, et al. Expression of enzymes and receptors of leukotriene pathway genes in equine endometrium during the estrous cycle and early pregnancy. Theriogenology. 2013;80(2):145–152. doi:10.1016/j.theriogenology.2013.03.025
  15. Atli MO, Guzeloglu A, Dinc DA. Expression of wingless type (WNT) genes and their antagonists at mRNA levels in equine endometrium during the estrous cycle and early pregnancy. Anim Reprod Sci. 2011;125(1-4):94–102. doi:10.1016/j.anireprosci.2011.04.001
  16. Atli MO, Kurar E, Kayis SA, et al. Evaluation of genes involved in prostaglandin action in equine endometrium during estrous cycle and early pregnancy. Anim Reprod Sci. 2010;122(1-2):124–132. doi:10.1016/j.anireprosci.2010.08.007
  17. Kayis SA, Atli MO, Kurar E, et al. Rating of putative housekeeping genes for quantitative gene expression analysis in cyclic and early pregnant equine endometrium. Anim Reprod Sci. 2011;125(1-4):124–132. doi:10.1016/j.anireprosci.2011.02.019
  18. Bołzan, Emilia; Andronowska, Aneta; Bodek, Gabriel; Morawska-Pucińska, Ewa; Krawczyński, Kamil; Dąbrowski, Adam; Zięcik, Adam J. The novel effect of hCG administration on luteal function maintenance during the estrous cycle/pregnancy and early embryo development in the pig. Polish Journal of Veterinary Sciences 2013, 16(2): 323-332 (doi: 10.2478/pjvs-2013-0044).
  19. Zięcik, Adam J.; Przygrodzka, Emilia; Kaczmarek, Monika M. Corpus luteum regression and early pregnancy maintenance in pigs. W: The life cycle of the Corpus Luteum. Ed.: Meidan, Rina. Springer International Publishing, Switzerland 2017 (ISBN 978-3-319-43236-6), Chapter 12: 227-248 (doi: 10.1007/978-3-319-43238-0_12).
  20. Kaczmarek, Monika M.; Krawczyński, Kamil; Najmuła, Joanna; Reliszko, Żaneta P.; Sikora, Małgorzata; Gajewski, Zdzisław. Differential expression of genes linked to the leukemia inhibitor factor signaling pathway during the estrus cycle and early pregnancy in the porcine endometrium. Reproductive Biology 2014, 14(4): 293-297 (doi: 10.1016/j.repbio.2014.06.003).
  21. Markiewicz, W., Bogacki, M., Blitek, M. et al. Comparison of the porcine uterine smooth muscle contractility on days 12–14 of the estrous cycle and pregnancy. Acta Vet Scand 58, 20 (2015). https://doi.org/10.1186/s13028-016-0201-z
  22. Kamińska K, Wasielak M, Bogacka I, Blitek M, Bogacki M. Quantitative expression of lysophosphatidic acid receptor 3 gene in porcine endometrium during the periimplantation period and estrous cycle. Prostaglandins Other Lipid Mediat. 2008;85(1-2):26–32. doi:10.1016/j.prostaglandins.2007.10.001
  23. Kowalik MK, Rekawiecki R, Kotwica J. Expression of membrane progestin receptors (mPRs) in the bovine corpus luteum during the estrous cycle and first trimester of pregnancy. Domest Anim Endocrinol. 2018;63:69–76. doi:10.1016/j.domaniend.2017.12.004
  24. Kimmins S, MacLaren LA. Oestrous cycle and pregnancy effects on the distribution of oestrogen and progesterone receptors in bovine endometrium. Placenta. 2001;22(8-9):742–748. doi:10.1053/plac.2001.0708
  25. Zavy MT, Bazer FW, Thatcher WW, Wilcox CJ. A study of prostaglandin F2 alpha as the luteolysin in swine: V. Comparison of prostaglandin F, progestins, estrone and estradiol in uterine flushings from pregnant and nonpregnant gilts. Prostaglandins. 1980;20(5):837–851. doi:10.1016/0090-6980(80)90137-9
  26. Frank M, Bazer FW, Thatcher WW, Wilcox CJ. A study of prostaglandin F2alpha as the luteolysin in swine: III effects of estradiol valerate on prostaglandin F, progestins, estrone and estradiol concentrations in the utero-ovarian vein of nonpregnant gilts. Prostaglandins. 1977;14(6):1183–1196. doi:10.1016/0090-6980(77)90295-7
  27. Moeljono MP, Thatcher WW, Bazer FW, Frank M, Owens LJ, Wilcox CJ. A study of prostaglandin F2alpha as the luteolysin in swine: II Characterization and comparison of prostaglandin F, estrogens and progestin concentrations in utero-ovarian vein plasma of nonpregnant and pregnant gilts. Prostaglandins. 1977;14(3):543–555. doi:10.1016/0090-6980(77)90268-4
  28. Basu S, Kindahl H. Prostaglandin biosynthesis and its regulation in the bovine endometrium: A comparison between nonpregnant and pregnant status. Theriogenology. 1987;28(2):175–193. doi:10.1016/0093-691x(87)90265-2
  29. Killeen AP, Diskin MG, Morris DG, Kenny DA, Waters SM. Endometrial gene expression in high- and low-fertility heifers in the late luteal phase of the estrous cycle and a comparison with midluteal gene expression. Physiol Genomics. 2016;48(4):306–319. doi:10.1152/physiolgenomics.00042.2015

Reviewer 2 Report

In this manuscript, Tanski and colleagues use quantitative RT-PCR to determine the relative levels of the mRNA of 3 aquaporin genes in various parts of the porcine oviduct, at various times during the estrous cycle, and during pregnancy.

The study extends previous observations made by the authors. The work is properly carried out but the presentation of the data is either unclear, as outlined below, or, worse, the work suffers from reproducibility issues that may weaken the conclusions. In the end, this study is descriptive and the evidence correlative, but it could form the basis for interesting functional studies.

The paper is generally well-written with a few minor edits remaining to be made.

Major comments:

The data shown in Figure 3 are very unclear. First the header “marked with a capital letter” (ESTROUS CYCLE/PREGNANCY) is unclear or misleading and does not match what is said in the figure legend.  The authors used a bit too much of “copy-paste” for the preparation of their legends. Then if, I interpret the data properly (I hope I don’t), there seems to be major inconsistencies between the data shown in this figure and those  shown in the earlier figures. Therefore, the authors must clearly explain what are the data shown in this figure and how they compare with the earlier figures.

The same comments apply to the data shown in Figure 4. I am getting the impression the experiments shown in the various figures were done independently. However, the relative levels seem to change between figures and the conclusions made in figures 1 and 2 may not be valid if they cannot be reproduced in figures 3 and 4.

Minor comments:

Line 122: “was” instead of “were”

In the first paragraph of the Discussion (lines  163 -172), the authors should refer  specifically to the appropriate figures when they make a statement, especially considering the major comments made above.

The last sentence of the conclusion is poorly built and must be rewritten.

Author Response

Response to Reviewer 2 Comments

We are sending the revised manuscript of our paper entitled “Changes in aquaporin 1, 5 and 9 gene expression in the pig oviduct according to estrous cycle and early pregnancy.” with hope it will fulfil scientific requirements to be published in your Journal

Point 1: The data shown in Figure 3 are very unclear. First the header “marked with a capital letter” (ESTROUS CYCLE/PREGNANCY) is unclear or misleading and does not match what is said in the figure legend.  The authors used a bit too much of “copy-paste” for the preparation of their legends. Then if, I interpret the data properly (I hope I don’t), there seems to be major inconsistencies between the data shown in this figure and those shown in the earlier figures. Therefore, the authors must clearly explain what are the data shown in this figure and how they compare with the earlier figures.

Response 1: We have changed and improved the legends of the graphs (Figure 1, 2, 3) as suggested.

Although every Reviewers opinion is appreciated we do not partly agree with this one.

We removed Fig 4 from the manuscript. However, data in Fig 3 is very important to show the comparision between  the luteal phase of the estrous cycle and early pregnancy, periods when corpus luteum is active. This model of comparison is well known in literature and often used by authors.

Below, there are several, our and others, peer reviewed publication that such comparison was made:

  1. Skowronski MT, Skowronska A, Nielsen S. Fluctuation of aquaporin 1, 5, and 9 expression in the pig oviduct during the estrous cycle and early pregnancy. J Histochem Cytochem. 2011;59(4):419–427. doi:10.1369/0022155411400874
  2. Skowronski MT. Distribution and quantitative changes in amounts of aquaporin 1, 5 and 9 in the pig uterus during the estrous cycle and early pregnancy. Reprod Biol Endocrinol. 2010;8:109. Published 2010 Sep 9. doi:10.1186/1477-7827-8-109
  3. Skowronska A, Mlotkowska P, Eliszewski M, Nielsen S, Skowronski MT. Expression of aquaporin 1, 5 and 9 in the ovarian follicles of cycling and early pregnant pigs. Physiol Res. 2015;64(2):237–245.
  4. Smolinska, N.; Kiezun, M.; Dobrzyn, K.; Rytelewska, E.; Kisielewska, K.; Gudelska, M.; Zaobidna, E.; Bogus-Nowakowska, K.; Wyrebek, J.; Bors, K.; Kopij, G.; Kaminska, B.; Kaminski, T. Expression of Chemerin and Its Receptors in the Porcine Hypothalamus and Plasma Chemerin Levels during the Oestrous Cycle and Early Pregnancy.  J. Mol. Sci.201920, 3887
  5. Kaczynski P, Bauersachs S, Baryla M, et al. Estradiol-17β-Induced Changes in the Porcine Endometrial Transcriptome In Vivo. Int J Mol Sci. 2020;21(3):E890. Published 2020 Jan 30. doi:10.3390/ijms21030890
  6. Smolinska N, Kaminski T, Siawrys G, Przala J. Expression of leptin and its receptor genes in the ovarian follicles of cycling and early pregnant pigs. Animal. 2013;7(1):109–117. doi:10.1017/S1751731112001103
  7. Wojciechowicz B, Kotwica G, Kolakowska J, Franczak A. The activity and localization of 3β-hydroxysteroid dehydrogenase/Δ(5)-Δ(4) isomerase and release of androstenedione and progesterone by uterine tissues during early pregnancy and the estrous cycle in pigs. J Reprod Dev. 2013;59(1):49–58. doi:10.1262/jrd.2012-099
  8. Franczak A, Wojciechowicz B, Kotwica G. Transcriptomic analysis of the porcine endometrium during early pregnancy and the estrous cycle. Reprod Biol. 2013;13(3):229–237. doi:10.1016/j.repbio.2013.07.001
  9. Klainbart S, Slon A, Kelmer E, et al. Global hemostasis in healthy bitches during pregnancy and at different estrous cycle stages: Evaluation of routine hemostatic tests and thromboelastometry. Theriogenology. 2017;97:57–66. doi:10.1016/j.theriogenology.2017.04.023
  10. Niswender GD, Juengel JL, McGuire WJ, Belfiore CJ, Wiltbank MC. Luteal function: the estrous cycle and early pregnancy. Biol Reprod. 1994;50(2):239–247. doi:10.1095/biolreprod50.2.239
  11. Kiewisz, Jolanta; Krawczyński, Kamil; Lisowski, Paweł; Blitek, Agnieszka; Zwierzchowski, Lech; Zięcik, Adam J.; Kaczmarek, Monika M. Global gene expression profiling of porcine endometria on days 12 and 16 of the estrous cycle and pregnancy. Theriogenology 2014, 82(6): 897-909 (doi: 10.1016/j.theriogenology.2014.07.009).
  12. Przygrodzka E, Witek KJ, Kaczmarek MM, Andronowska A, Ziecik AJ. Expression of factors associated with apoptosis in the porcine corpus luteum throughout the luteal phase of the estrous cycle and early pregnancy: their possible involvement in acquisition of luteolytic sensitivity. Theriogenology. 2015;83(4):535–545. doi:10.1016/j.theriogenology.2014.10.016
  13. Korzekwa AJ, Bah MM, Kurzynowski A, Lukasik K, Groblewska A, Skarzynski DJ. Leukotrienes modulate secretion of progesterone and prostaglandins during the estrous cycle and early pregnancy in cattle: an in vivo study. Reproduction. 2010;140(5):767–776. doi:10.1530/REP-10-0202
  14. Guzeloglu A, Atli MO, Kurar E, et al. Expression of enzymes and receptors of leukotriene pathway genes in equine endometrium during the estrous cycle and early pregnancy. Theriogenology. 2013;80(2):145–152. doi:10.1016/j.theriogenology.2013.03.025
  15. Atli MO, Guzeloglu A, Dinc DA. Expression of wingless type (WNT) genes and their antagonists at mRNA levels in equine endometrium during the estrous cycle and early pregnancy. Anim Reprod Sci. 2011;125(1-4):94–102. doi:10.1016/j.anireprosci.2011.04.001
  16. Atli MO, Kurar E, Kayis SA, et al. Evaluation of genes involved in prostaglandin action in equine endometrium during estrous cycle and early pregnancy. Anim Reprod Sci. 2010;122(1-2):124–132. doi:10.1016/j.anireprosci.2010.08.007
  17. Kayis SA, Atli MO, Kurar E, et al. Rating of putative housekeeping genes for quantitative gene expression analysis in cyclic and early pregnant equine endometrium. Anim Reprod Sci. 2011;125(1-4):124–132. doi:10.1016/j.anireprosci.2011.02.019
  18. Bołzan, Emilia; Andronowska, Aneta; Bodek, Gabriel; Morawska-Pucińska, Ewa; Krawczyński, Kamil; Dąbrowski, Adam; Zięcik, Adam J. The novel effect of hCG administration on luteal function maintenance during the estrous cycle/pregnancy and early embryo development in the pig. Polish Journal of Veterinary Sciences 2013, 16(2): 323-332 (doi: 10.2478/pjvs-2013-0044).
  19. Zięcik, Adam J.; Przygrodzka, Emilia; Kaczmarek, Monika M. Corpus luteum regression and early pregnancy maintenance in pigs. W: The life cycle of the Corpus Luteum. Ed.: Meidan, Rina. Springer International Publishing, Switzerland 2017 (ISBN 978-3-319-43236-6), Chapter 12: 227-248 (doi: 10.1007/978-3-319-43238-0_12).
  20. Kaczmarek, Monika M.; Krawczyński, Kamil; Najmuła, Joanna; Reliszko, Żaneta P.; Sikora, Małgorzata; Gajewski, Zdzisław. Differential expression of genes linked to the leukemia inhibitor factor signaling pathway during the estrus cycle and early pregnancy in the porcine endometrium. Reproductive Biology 2014, 14(4): 293-297 (doi: 10.1016/j.repbio.2014.06.003).
  21. Markiewicz, W., Bogacki, M., Blitek, M. et al. Comparison of the porcine uterine smooth muscle contractility on days 12–14 of the estrous cycle and pregnancy. Acta Vet Scand 58, 20 (2015). https://doi.org/10.1186/s13028-016-0201-z
  22. Kamińska K, Wasielak M, Bogacka I, Blitek M, Bogacki M. Quantitative expression of lysophosphatidic acid receptor 3 gene in porcine endometrium during the periimplantation period and estrous cycle. Prostaglandins Other Lipid Mediat. 2008;85(1-2):26–32. doi:10.1016/j.prostaglandins.2007.10.001
  23. Kowalik MK, Rekawiecki R, Kotwica J. Expression of membrane progestin receptors (mPRs) in the bovine corpus luteum during the estrous cycle and first trimester of pregnancy. Domest Anim Endocrinol. 2018;63:69–76. doi:10.1016/j.domaniend.2017.12.004
  24. Kimmins S, MacLaren LA. Oestrous cycle and pregnancy effects on the distribution of oestrogen and progesterone receptors in bovine endometrium. Placenta. 2001;22(8-9):742–748. doi:10.1053/plac.2001.0708
  25. Zavy MT, Bazer FW, Thatcher WW, Wilcox CJ. A study of prostaglandin F2 alpha as the luteolysin in swine: V. Comparison of prostaglandin F, progestins, estrone and estradiol in uterine flushings from pregnant and nonpregnant gilts. Prostaglandins. 1980;20(5):837–851. doi:10.1016/0090-6980(80)90137-9
  26. Frank M, Bazer FW, Thatcher WW, Wilcox CJ. A study of prostaglandin F2alpha as the luteolysin in swine: III effects of estradiol valerate on prostaglandin F, progestins, estrone and estradiol concentrations in the utero-ovarian vein of nonpregnant gilts. Prostaglandins. 1977;14(6):1183–1196. doi:10.1016/0090-6980(77)90295-7
  27. Moeljono MP, Thatcher WW, Bazer FW, Frank M, Owens LJ, Wilcox CJ. A study of prostaglandin F2alpha as the luteolysin in swine: II Characterization and comparison of prostaglandin F, estrogens and progestin concentrations in utero-ovarian vein plasma of nonpregnant and pregnant gilts. Prostaglandins. 1977;14(3):543–555. doi:10.1016/0090-6980(77)90268-4
  28. Basu S, Kindahl H. Prostaglandin biosynthesis and its regulation in the bovine endometrium: A comparison between nonpregnant and pregnant status. Theriogenology. 1987;28(2):175–193. doi:10.1016/0093-691x(87)90265-2
  29. Killeen AP, Diskin MG, Morris DG, Kenny DA, Waters SM. Endometrial gene expression in high- and low-fertility heifers in the late luteal phase of the estrous cycle and a comparison with midluteal gene expression. Physiol Genomics. 2016;48(4):306–319. doi:10.1152/physiolgenomics.00042.2015

Point 2: The same comments apply to the data shown in Figure 4. I am getting the impression the experiments shown in the various figures were done independently. However, the relative levels seem to change between figures and the conclusions made in figures 1 and 2 may not be valid if they cannot be reproduced in figures 3 and 4.

Response 2: During the preparation of the manuscript an error crept in. We are very sorry for that. We changed Figure 1C and 3A. All figures are correct and attached.  

Point 3: Line 122: “was” instead of “were”

Response 3: Corrected.

Point 4: In the first paragraph of the Discussion (lines  163 -172), the authors should refer  specifically to the appropriate figures when they make a statement, especially considering the major comments made above.

Response 4: We changed the Discussion part as suggested.

Point 5: The last sentence of the conclusion is poorly built and must be rewritten.

Response 5: Thank you for your attention regarding the Conclusion. We amended the conclusion part.

Round 2

Reviewer 2 Report

The authors adequately addressed the concerns I had with theoriginal version of the manuscript.

I only have very minor edits to the text:

Line 136 : « comparisons » instead of « comparized »

Line 137: ibid

Line 55 “compared”

Author Response

In response to Referees

The referees are thanked very much for helpful comments.

Response to Reviewer 2 Comments

Point 1: I only have very minor edits to the text:

Line 136 : « comparisons » instead of « comparized »

Line 137: ibid

Line 55 “compared

Response 1: We have changed all suggested words in text